# How to Shut Down Transcription in Archaea during Virus Infection

**DOI:** 10.3390/microorganisms10091824

**Published:** 2022-09-13

**Authors:** Simona Pilotto, Finn Werner

**Affiliations:** RNAP Laboratory, Institute for Structural and Molecular Biology, University College London, Gower Street, London WC1E 6BT, UK

**Keywords:** archaea, transcription inhibition, RNA polymerase, viruses, evolution, antibiotics

## Abstract

Multisubunit RNA polymerases (RNAPs) carry out transcription in all domains of life; during virus infection, RNAPs are targeted by transcription factors encoded by either the cell or the virus, resulting in the global repression of transcription with distinct outcomes for different host–virus combinations. These repressors serve as versatile molecular probes to study RNAP mechanisms, as well as aid the exploration of druggable sites for the development of new antibiotics. Here, we review the mechanisms and structural basis of RNAP inhibition by the viral repressor RIP and the crenarchaeal negative regulator TFS4, which follow distinct strategies. RIP operates by occluding the DNA-binding channel and mimicking the initiation factor TFB/TFIIB. RIP binds tightly to the clamp and locks it into one fixed position, thereby preventing conformational oscillations that are critical for RNAP function as it progresses through the transcription cycle. TFS4 engages with RNAP in a similar manner to transcript cleavage factors such as TFS/TFIIS through the NTP-entry channel; TFS4 interferes with the trigger loop and bridge helix within the active site by occlusion and allosteric mechanisms, respectively. The conformational changes in RNAP described above are universally conserved and are also seen in inactive dimers of eukaryotic RNAPI and several inhibited RNAP complexes of both bacterial and eukaryotic RNA polymerases, including inactive states that precede transcription termination. A comparison of target sites and inhibitory mechanisms reveals that proteinaceous repressors and RNAP-specific antibiotics use surprisingly common ways to inhibit RNAP function.

## 1. Introduction

Archaea originally rose to fame due the unorthodox environmental niches they dominated, including extremes of temperature, pressure, salinity and pH. Over the last decade, however, it has transpired that archaea are highly abundant organisms in the biosphere, and many species thrive under moderate conditions on a par with mesophilic bacteria. As an integral part of the biosphere in the Earth’s crust and oceans, archaea play an important role in the elemental cycles including the recycling of sulphur and nitrogen [1]. While archaea can be found in the human microbiome, in particular, the gut and mouth, it remains a mystery why no severe human disease has been associated with an archaeal pathogen to date [2,3,4]. Perhaps one of the most intriguing aspects of archaea is their role in eukaryogenesis; despite an on-going debate, it is now widely accepted by most researchers in the field that archaea belonging to the Asgård phylum are the direct ancestors of eukaryotes [5]. In lieu of this position in the universal tree of life, the molecular machines that carry out the processing of biological information, including replication, transcription and translation, are closely related to their eukaryotic counterparts.

### 1.1. The Archaeal Transcription Machinery

The archaeal genomes are organised in polycistronic operons, such as in bacteria, and compacted using a range of nucleoid proteins that are either specific for archaea (like Cren7 and Alba) or eukaryotes (histones) [6,7]. The latter are comprised on the canonical histone fold and form so-called hyper-nucleosomes where histone dimers oligomerize to form a helical ramp around which the DNA wraps as a solenoid [8]. The archaeal RNA polymerase is made of subunits that are homologous to eukaryotic RNAPII, including RPB1-12 (named Rpo1-12) with the exception of RPB9, and minor variations in different archaeal phyla [7]. These include split RPB1 and 2 subunits and the absence of RPB/Rpo8 homologues in some species, and the addition of the only archaea-specific subunit Rpo13 in others. While RPB9 does not seem to be present throughout, the paralogous transcript cleavage factor TFS likely fulfils an analogous function and is widely conserved in archaea [9,10]. The archaeal RNAP is supported by ‘general’ transcription initiation factors that are homologous to eukaryotic transcription systems, including the TATA-binding protein TBP, TFIIB (TFB in archaea) and TFIIE (TFE) [7,9,11]. Archaeal promoters are likewise characterised by eukaryote-like sequence motifs including the TATA-box, B recognition element (BRE) and Initiator (Inr) that interact with TBP, TFIIB and the RNAP, respectively [12,13]. The sequential assembly of TBP, TFB and RNAP on the promoter forms the ‘minimal’ transcription preinitiation complex (PIC), which also corresponds to the combination of factors that is necessary and sufficient to enable RNAPII transcription using strong promoters and negatively supercoiled templates [14]. TFE joins the complex and enhances DNA melting, forming the complete archaeal PIC [13,15]. Like all RNAPs, the archaeal enzyme makes repeated use of the DNA template by progressing through three distinct phases of the transcription cycle: initiation, elongation and termination (Figure 1). 

Following the escape of RNAP from the promoter, the elongation factors Spt4/5 and Elf1 are recruited to the transcription elongation complex (TEC) and improve the processivity of elongation by sealing the DNA-binding channel and forming a tunnel through which the downstream DNA is directed to the catalytic centre of RNAP [16]. Termination of transcription is achieved by factor-dependent and intrinsic mechanisms, the former reliant on the archaeal cleavage and polyadenylation specificity factor 1 (aCPSF1), homologous of the eukaryotic CPSF73, and the latter enabled by short poly-U stretches in the RNA transcript [17,18]. Nonbiased whole-genome occupancy studies have revealed that the termination factor is recruited to promoter-proximal locations in addition to the 3′ regions of transcription units. The occupancy of aCPSF1 anticorrelates with mRNA levels genome-wide, suggesting that transcription in archaea is regulated by promoter-proximal premature termination [12].

### 1.2. Viruses versus Hosts

Viruses and their hosts are engaged in an endless battle, whose outcome is called by the arms race between host and viral measures and countermeasures [19]. Yet, in many cases, viruses thrive in a coexistence with their host (commensalism), and it is now well accepted that viruses play a key role for horizontal gene transfer and are drivers of evolution of life [20,21]. In bacteria and eukaryotes, virus–host pairs where viruses encode their own RNA polymerase, which include many bacteriophages and metazoan viruses, viral factors quickly and specifically shut down the host’s gene expression to favor the viral transcription program [22,23]. This redirects host resources to produce virions and limits the host’s ability to mount an effective immune response to the infection. Conversely, the host in many cases attenuates its own transcription as part of the early response to sudden unfavourable growth conditions and enters a quiescent state, which can provide the infected cell with the opportunity to mount additional antiviral responses, or alternatively as an initial step leading to cell death aiming to limit virus propagation in the community [24]. The mechanisms underlying RNAP repression differ between viral and cellular transcription systems in bacteria, archaea and eukarya, depending on the complexity and regulatory strategies of each system.

### 1.3. Virus-Encoded Inhibitors

Viral gene expression is typically regulated temporally (early–middle–late) as a function of the infection time course, in contrast to cellular gene expression that responds to complex environmental or developmental cues. As a rule, the early genes of bacteriophages are under the control of host-like promoters and are expressed by the host RNAP [25]. Early gene products include the phage RNAP and inhibitors of the host RNAP. Once a critical threshold of both factors is reached, a switch from early to late transcription occurs, concomitantly with the effective repression of all host genes [23,26]. Variations to the described mechanism include phages that encode for two RNAPs that enable a more complex phage gene expression [27]. The *coliphage* T7 encodes Gp2, a small protein that binds the RNAP-σ70 holoenzyme inside the DNA-binding channel, which inhibits the open complex formation [23,26,28]. Instead, other viral factors modulate post-translational modifications of RNAP subunits, which regulate RNAP activity. The eukaryotic *herpes simplex* virus-1 factor ICP22 (infected cell protein 22) [22] and the *Bunyamwera* virus factor NSs (non-structural protein S) interfere with the phosphorylation of the RNA polymerase II subunit RPB1 [29], while the Togaviridae factor nsP2 promotes RPB1 ubiquitination, which leads to a rapid depletion of RNAPII [30]. Viruses that do not encode their own RNAP are dependent on the host RNAP and, therefore, they tend to deploy regulatory factors that interfere with host promoter recognition, aiming to redistribute RNAPs from host to viral promoters. The recently sequenced Italian rudiviruses encode homologues of the initiation factor TFB, but it has not been established whether the viral TFBs (vTFB) preferentially direct transcription from viral compared to cellular promoters [31].

### 1.4. Cellular Inhibitors

Separate from the virus–host scenario, cells utilise global negative regulation of transcription via RNAP and transcription factors. Several negative regulators of bacterial RNAP have been described in the literature; these are often expressed or activated during unfavourable growth conditions, such as low pH for *Thermus thermophilus* Gfh1 (Gre factor homologue 1) [32,33], or nutrient deficiency for *Escherichia coli* DksA (DnaK suppressor A) [34,35]. Likewise, eukaryotic transcription is also regulated on a global level. Naïve lymphocytes are transcriptionally silent on a genome-wide level due to severely repressed levels of the initiation factor TFIIH, a factor that catalyzes DNA strand separation and template loading into the RNAP active site during initiation [36]. Lymphocyte activation triggers TFIIH expression, which, in turn, results in rapid global amplification of the RNAPII transcriptome. The negative elongation factor (NELF) enables promoter-proximal pausing of RNAPII, and plays an important role in viral latency during HIV (human immunodeficiency virus) infection [37,38]. Finally, MAF1, whose expression is triggered by nutritional deficit and other cellular stress conditions, binds directly to the clamp of RNAPIII and inhibits it, while also interacting with mTOR and thereby indirectly activating the host immune response [39,40,41,42].

The genomes of more than 230 archaeal viruses have been sequenced, all of them are DNA-based, and none of them encode genes with homology to any RNA polymerase subunit (with the exception of primase) [43]. Transcription in archaeal viruses is poorly understood, but in agreement with the absence of viral RNAPs, viral promoters include TATA and BRE host promoter consensus elements [44]. As such, they recruit the host-encoded initiation factors TBP and TFB to facilitate transcription by the host RNAP [45], which has important implications, as repressing host RNAP would also affect virus transcription. So far, few viral proteins have been described as interfering with transcription, they contain the ribbon–helix–helix (RHH) fold, which is typical of DNA binding proteins in bacteria and archaea [46]. They act as repressors by binding to specific DNA sequences, mainly as dimers. The targets of these repressors are viral genes that require a specific time window for the expression, and none of them have been shown to interfere with host transcription. However, viral infection in archaea, as well as in the other domains of life, cause a strong downregulation of host transcription, which can only be explained as the result of yet to be discovered repressors expressed by either the virus redirecting host resources or the host aiming to fight the virus. In this review, we focus on the only two known direct RNAP inhibitors, which play a key role in the virus–host conflict in archaea, the *Acidianus* two-tailed virus RNAP inhibitory protein, ATV RIP and the *Saccharolobus solfataricus* (Sso) negative regulator TFS4 [10,44,45,46].

## 2. RIP Forms a Plug in the DNA-Binding Channel

ATV is a double-stranded DNA virus that infects hyperthermophiles of the Sulfolobaceae family, including *Acidianus* and *Saccharolobus* species [47]. The ATV ORF145 gene product called RIP (RNAP inhibitory protein) is evolutionary related to the viral structural protein P131 (ORF131) sharing 36% sequence identity; however, the protein has undergone an intriguing specialisation. The result is a protein that forms a high-affinity complex with RNAP, effectively inhibiting transcription initiation and elongation [44] (Figure 1).

RIP is a protein with a compact six-helical bundle, which is highly similar to P131, but in addition, RIP includes a long C-terminal tail that is not conserved in P131 or homologues in other viruses. The high-resolution cryo-EM structure of RIP bound to the archaeal RNAP (pdb code 7oq4) reveals that the helical bundle forms a plug in the DNA-binding channel, while the C-terminal tail runs along the clamp core enabling the high affinity binding to RNAP (Figure 2a) [46].

The clamp is a flexible domain, which opens and closes over the channel upon DNA binding and in response to transcription factors as RNAP progresses through the transcription cycle [49,50]. Single molecule FRET studies that monitor clamp opening and closing movements show that RIP binding locks the clamp into one fixed position [46]. As RIP seals off the DNA-binding channel, the RNAP–RIP complex cannot engage with the promoter DNA during initiation. In addition, RIP binding to the clamp sterically occludes RNAP interactions with the initiation factor TFB and the elongation factors Spt4/5, of which each also compete for binding to RNAP. A deeper structural analysis and comparison with the eukaryotic PIC show that helix-4 and the C-terminal tail of RIP topologically mimic the B-linker of TFIIB, homologous of archaeal TFB (Figure 2b) [46,48]. RIP is perhaps one of the most extreme examples of evolutionary diversification where the common ancestor of P131 and RIP underwent gene duplication and speciation. Over time, one of the two paralogues evolved into a structural component of the virus, while the other acquired a C-terminal extension, which enabled RIP to bind and ultimately inhibit RNAP.

RIP expression is induced during the late stages of infection prior to cell lysis [44,46]. The strict time window of RIP expression is compatible with a role in virus particle assembly. The number of viral genomes is high during this stage, and as they are actively transcribed to produce viral proteins, they are most likely also adorned with RNAPs. The high occupancy of RNAP, in turn, is likely to be counterproductive during the packaging of viral DNA into the tight environment of the virus particle. RIP could be addressing this challenge by stripping RNAP off the viral genomes. In addition, the concomitant shutoff of the host transcription can activate cell death pathways leading to the lytic event and virions release.

## 3. TFS4 Uses a ‘Belts and Braces’ Approach to Inhibition

The transcript cleavage factors (TFS) are a large family of transcription regulators that interact with RNAP through its NTP-entry channel. This channel consists of a pore at the bottom of a funnel that enables the access of nucleotides to the RNAP active site, and the egress of the RNA 3′-end in the backtracked TEC [51,52]. TFS factors utilise a zinc ribbon domain at their C-terminus to position two carboxylate side chains in the catalytic site of RNAP, thereby triggering the cleavage of the backtracked RNA, allowing transcription elongation to commence [10,51,52] (Figure 1). The importance of this cleavage function is highlighted by convergent evolution; while archaeal and eukaryotic cleavage factors are homologous, the bacterial Gre factors are not sequentially nor structurally related to TFS, nevertheless, they employ an identical mechanism of reactivating stalled TECs [53,54,55]. In addition, eukaryotes have stably incorporated TFIIS paralogues as RNAP subunits, RPB9 in RNAPII [52], RPA12 in RNAPI [56,57] and RPC10 in RNAPIII [58]. These RNAP subunits have acquired alternative functions that negatively influence elongation as both RPA12 and RPC10 are implicated in transcription termination. The function of RPB9 is unclear, it cannot stimulate cleavage but is essential for transcription-coupled DNA repair in yeast; curiously, the rpb9 deletion can be suppressed by the impairment of the elongation factor Spt4/5 function through the deletion of Spt4 [59].

Archaea encode only one type of RNAP, but several TFS paralogues [10], suggesting that each paralog has a distinct function. In Sulfolobales, TFS1 enhances transcript cleavage, rescuing stalled TECs and improving the elongation rate overall [10,51,60]. We have recently shown that the TFS4 paralogue, which lacks the two catalytic carboxylate residues characteristic of authentic cleavage factors, efficiently inhibits the archaeal RNAP [10,46]. The high-resolution cryo-EM map of the RNAP/TFS4 complex shows that TFS4 interacts with RNAP similarly to canonical cleavage factors, with the N-terminal zinc ribbon (ZR^N^) binding between the lobe and the upper jaw, and, through the linker, positioning the C-terminal zinc ribbon (ZR^C^) inside the NTP-entry channel (Figure 3a,b) [46].

However, as the resulting binding surface is not solvent-exposed, TFS4 displaces the jaw, which causes a large conformational change with concomitant widening of the DNA-binding channel and melting of the bridge helix (Figure 3c). Finally, the ZR^C^ of TFS4 physically clashes with and displaces the trigger loop. As the bridge helix and trigger loop are essential for the catalytic addition and translocation of NTPs during RNA synthesis, TFS4 effectively inhibits the enzymatic activity [10]. Moreover, the widening of the DNA-binding channel likely weakens RNAP–DNA interactions during elongation and interferes with PIC formation (Figure 1) [10].

RNAPs are large and dynamic molecular machines; the engagement with transcription factors, nucleic acids and the enzymatic reaction itself all involve conformational changes in RNAP, most of which are intrinsic to its structure and, therefore, fundamentally evolutionary conserved [7,50]. The substantial conformational changes induced by TFS4 can be easily recognised in other related inactive states, including the inhibition of bacterial RNAP by Gfh1 [33], the dimerization of inactive RNAPI [56], and transition states preceding termination of RNAPI and III, which involve the TFS4 paralogues RPA12 and RPC10 [61,62]. TFS1 and TFS4 are derived from a common ancestor, which following gene duplication, have undergone speciation and evolved into a positive elongation factor and an inhibitor of transcription, respectively. This turned a positive transcription factor into a potent RNAP inhibitor. The *Saccharolobus solfataricus* TFS4 paralogue is not expressed during standard cell growth conditions, either during the exponential or stationary phase, but 12 h after infection with the *Sulfolobus* turreted icosahedral virus (STIV), the protein becomes detectable and shortly after the cell growth stagnates. The expression of a TFS4 variant from a plasmid induces growth arrest of *Sulfolobus acidocaldarius*, making it tempting to speculate that TFS4 is the causative agent of the growth phenotype caused by STIV infection. A dormant state or quiescence is often associated with a persister state in bacteria [56], allowing the infected cell to mount additional immune responses to clear the virus.

## 4. Similarities between Antibiotics, Inhibitors, TFS4 and RIP

The wide-spread emergence of multidrug resistant bacterial pathogens calls for urgent efforts to discover and develop novel classes of antibiotics [63]. RNAPs are powerful targets to fight infectious diseases by inhibiting the pathogen’s RNAP, such as rifamycins, which only inhibit bacterial RNAPs [64], or nucleobase or NTP substrate analogues, which are generally used to target the viral RNA-dependent RNAPs [65]. Another therapeutical application of RNAP inhibition includes anticancer treatments, which either perturb the DNA template (e.g. cis-platinum or actinomycin D) [66], specifically target RNAPI [67] and RNAPII [68], or transcription factors that regulate RNAPII (e. g. p53 or myc) [69,70].

Antibiotics need to inhibit pathogen RNAPs with sufficient affinity and high selectivity to circumvent off-target effects, such as inhibiting the host RNAP and/or other macromolecules. In addition, when considering specific RNAP motifs to be drugged, it is important to consider the likely emergence of resistance [71]. One of the most well-characterised antibiotics is Rifampicin (Rif), which is widely used as a front-line drug in the treatment of tuberculosis. Rif does not bind the RNAP active site but binds in the RNA binding pocket and blocks RNA extension. Rif resistant strains emerge readily, and single amino acid substitutions in only three rpoB residues account for 88% of all clinically isolated Rif resistant tuberculosis strains [72]. For any antibiotic, the ideal target is small as well as critical for enzyme function, as this makes it less likely that mutations arise that are both (i) deficient in antibiotic binding and (ii) remain enzymatically active. Understanding the mechanistic and structural basis of RNAP inhibition is important to rationalise the action of known drugs and to design new and improved antibiotics. As the structure and function of RNAPs is evolutionary conserved, the insights obtained from studying the inhibition of the archaeal RNAP by RIP and TFS4 can contribute to understanding the molecular basis of RNAP inhibition, the identification of novel target sites and the search for new and improved inhibitors of bacterial and eukaryotic RNAPs. The field of RNAP-targeting antibiotics has been extensively reviewed elsewhere [71,73]. Below, we focus on a selection of antibiotics that interfere with RNAP reminiscent of RIP or TFS4 mechanisms of inhibition (Figure 4).

The structurally mobile motifs of RNAP include the active site bridge helix and trigger loop, and the RNAP clamp that moves on the so-called switches. While the continuous conformational changes in the bridge helix and trigger loop enable the catalytic activity and nucleotide translocation [81,82], the switch region controls the opening and closing movements of the clamp, which modulates the width of the DNA-binding channel, that is essential during DNA loading during initiation and for elongation complex stability [50,83]. These sites are powerful [84] drug targets, as exemplified by the discovery of numerous molecules of natural origin binding in those sites and efficiently inhibiting the RNA polymerase. The lasso peptide Microcin J25 binds in the NTP-entry channel, where it locks the bridge helix, blocking the access of NTP substrates to the active site and preventing trigger loop refolding [78]; Salinamide A binds to the bridge helix and interferes with conformational changes necessary for nucleotide addition [79], while Streptolydigin targets the bacterial RNAP bridge helix and trigger loop from the DNA-binding channel without interfering with NTP access [75]. In eukaryotes, the mushroom toxin alpha-amanitin traps the trigger loop and bridge helix in the active site impairing nucleotide incorporation and translocation of RNAPII [84,85]. Examples of antibiotics that freeze clamp movements include the approved drugs Fidaxomicin and Myxopyronin B, which bind to the switch region of bacterial RNAP [86]. The mechanisms underlying the inhibition of archaeal RNAP by RIP and TFS4 combine features of the above mechanisms. Similar to the switch-targeting antibiotic Myxopyronin B and Fidaxomicin, RIP abrogates the mobility of the RNAP clamp. Similar to Microcin J25, TFS4 binds in the NTP-entry channel, interfering with both the trigger loop and bridge helix. Furthermore, reminiscent of Fidaxomicin and alpha-amanitin, TFS4 induces a widening of the DNA-binding channel.

These compelling similarities demonstrate, that while many roads lead to Rome, travelers often share the same path.

## 5. Discussion

RNA polymerase inhibition plays an important role for the global regulation of gene expression in response to environmental changes, such as oxidative stress and nutrient deficiency, and in response to virus infection. Surprisingly, it appears that the global repression of transcription can be of both selective advantage for the host and the virus, but for entirely different reasons. These include improved survival for the host, and aiding virus particle assembly and release for the virus. As repressors often use a range of measures to inhibit RNAPs that are reminiscent of antibiotics, characterising the underlying inhibitory mechanisms is not only of academic interest but has the potential to benefit the development of more potent and specific antibiotics that are less prone to resistance.

### Take Home Messages

RNAPs and the molecular mechanisms of RNA synthesis are universally conserved in all domains of life.Both the virus and host cell can encode repressors that tightly bind to RNAPs and efficiently inhibit their functions, leading to transcriptome repression or attenuation.The *modus operandi* of these factors can unravel the underlying molecular mechanisms of RNAP activity and identify critical pressure points of enzyme functionInhibitory mechanisms include:
○steric occlusion of DNA, NTPs and transcription factor binding sites;○allosteric regulation by inducing conformational changes that perturb the active site (bridge helix and trigger loop) and widening of the DNA-binding channel.Allosteric mechanisms of inhibition can reveal movements inherent to RNAP function. As these are evolutionary conserved, it is possible to draw intriguing parallels between the regulation of different transcription systems.Antibiotics that target and inhibit RNAPs and proteinaceous repressors act via functionally closely related molecular mechanisms.A thorough understanding of RNAP inhibition in all domains of life, including archaea, could be beneficial for the development of novel antibiotics.

## Figures and Tables

**Figure 1 microorganisms-10-01824-f001:**
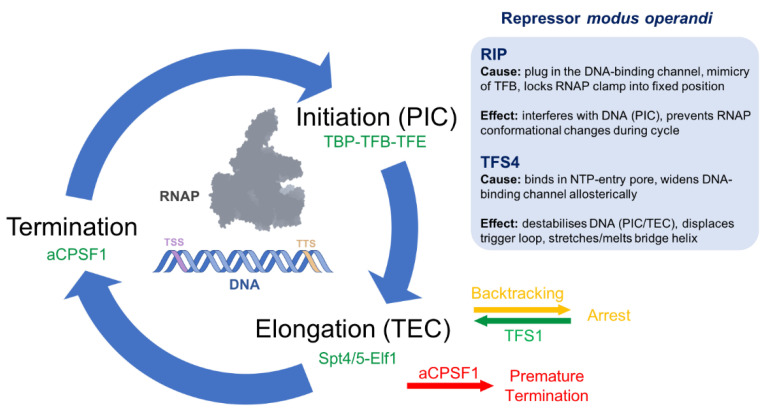
The archaeal transcription cycle. TBP, TFB and TFE form together with RNAP, the PIC on the archaeal promoter, to initiate transcription on the transcription start site (TSS). Following promoter escape, the elongation factors Spt4/5 and Elf1 associate with the RNAP forming the TEC. When encountering roadblocks, the TEC pauses, and the RNA is backtracked through the NTP-entry pore. TFS1 promotes the cleavage of the ‘excess’ of RNA in the pore, creates a new RNA 3′-end, which reactivates the TEC. Transcription termination can occur through a factor-independent or -dependent mechanisms at the transcription termination site (TTS) and utilises aCPSF1. aCPSF1 is also recruited proximal to the promoter where it can lead to premature termination. The inhibitory regulators RIP and TFS4 interfere with RNAP as it progresses through the transcription cycle as listed in the light blue box.

**Figure 2 microorganisms-10-01824-f002:**
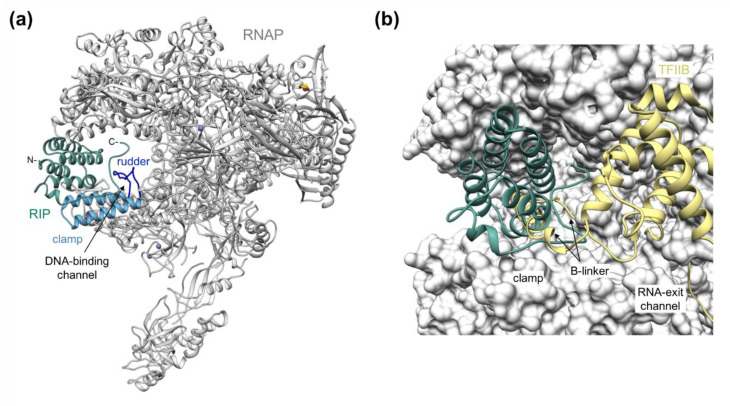
The archaeal RNAP–RIP complex. (**a**) The *Sulfolobus acidocaldarius* RNAP is shown in ribbon style with ions as spheres, zinc in medium purple, iron in red, and sulfur in yellow. RIP is highlighted in sea green with the N- and C-termini annotated, the clamp in steel blue, and the rudder in blue (pdb 7oq4). (**b**) Superimposition of the RNAP–RIP complex with TFIIB of the eukaryotic PIC (pdb 6gyk [48]). The archaeal RNAP is shown as surface with RIP highlighted in sea green as ribbon style, while for the eukaryotic PIC complex only TFIIB has been shown (in yellow).

**Figure 3 microorganisms-10-01824-f003:**
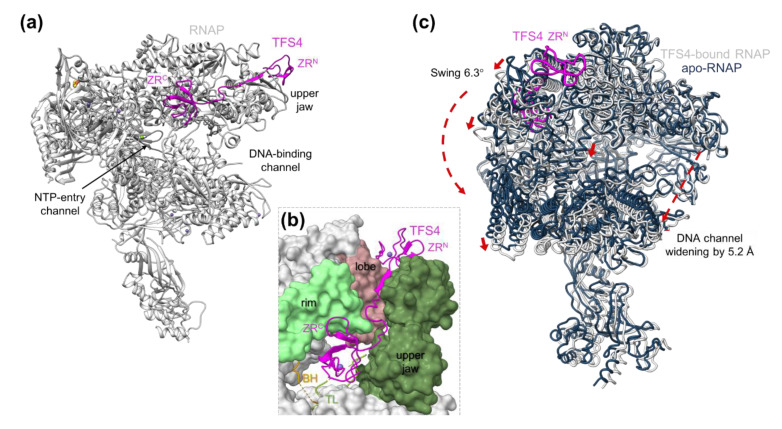
Molecular mechanisms of TFS4 inhibition. (**a**) Cryo-EM structure of the RNAP–TFS4 complex (pdb 7oqy) in ribbon style with ions as spheres, zinc in medium purple, magnesium in green, iron in red, and sulphur in yellow. TFS4 is shown in magenta with the N- and C-terminal zinc ribbon domains indicated. (**b**) Enlargement of the TFS4 binding site shown as surface with the lobe in dark pink, the upper jaw in olive, and the rim helices in light green, while bridge helix, BH in gold, trigger loop, TL in lime, and TFS4 are in ribbon style. Zinc ions are displayed as spheres in medium purple. (**c**) Superimposition of the TFS4–RNAP complex (in grey) with the apo-RNAP (in dark blue from pdb 7ok0). The red arrows indicate the direction of the downwards swinging of the jaw and clamp, as well as the stretch of the bridge helix and the extent of the DNA-binding channel opening.

**Figure 4 microorganisms-10-01824-f004:**
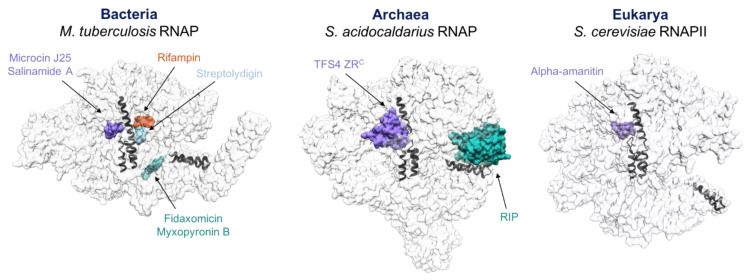
RNAP-specific antibiotics that are reminiscent of RIP and TFS4 action. Schematic illustration of the RNAP target sites for a selection of antibiotics and inhibitors. The RNAPs are shown as grey surfaces in transparency, the antibiotics and the inhibitors TFS4 and RIP are shown as surface representation using a colour code based on closely related binding sites: medium purple for the NTP-entry channel, orange for the RNA binding site, light sky blue for the BH/TL site, and sea green for the clamp region. All bacterial structures were superimposed against subunit beta; pdb codes: 5uh6 (Rifampin) [74], 2a6h (Streptolydigin) [75], 4yfx (Myxopyronin B) [76], 6fbv (Fidaxomicin) [77], 6n60 (Microcin J25) [78], 4mex (Salinamide A) [79]. *S. cerevisiae* RNAPII was obtained from pdb code 3cqz [80]. For clarity, only one compound for target site is shown.

## Data Availability

Publicly available data were analyzed in this study. This data can be found here: [https://www.rcsb.org/]; accession numbers are reported in the main text or in the figure legends for each structure used.

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
