# Peer review of "How to Shut Down Transcription in Archaea during Virus Infection"

_microorganisms, 2022, doi:10.3390/microorganisms10091824_

Round 1

Reviewer 1 Report

The manuscript by Pilotto and Werner is an excellent review article, well written, and highly informative. The authors did an excellent job explaining why findings in the field of archaea are highly relevant to biology in general.

This review is about archaea; therefore, moving the introduction on archaea and archaeal transcription earlier in the text would improve comprehension, especially for readers not familiar with the molecular biology of archaea. While this review focuses on RIP and TFS4, it would be extremely interesting to know whether RNAP inhibitors/negative regulators have been found in other archaeal viruses/archaea.

Figures: Please use “simple colors” rather than “sea green” and explain what is on the figs ( i.e., rudder) and the color scheme for each;  in Fig.2 B, eTFIIB looks yellow to me; in Figs 2 and 3, colored ions are not visible for the most part; in Fig. 4, it is difficult to see the different colors for Myxopyronin and Fidaxomicin.

Specific comments:

L 50: remove one “reached”

L54-56: awkward sentence; please rewrite

L79: give a bit more info about MAF1; this comes out of the blue after the sentence on HIV.

L83-86: moves “,” so it is easier to understand, i.e., “The archaeal transcription cycle is closely related to RNAPII that, in addition to the RNAPII subunits includes bona fide homologues of general transcription factors TBP, TFIIB, TFIIE, TFIIS, Spt4/5, Elf1, and CPSF73…

Author Response

This review is about archaea; therefore, moving the introduction on archaea and archaeal transcription earlier in the text would improve comprehension, especially for readers not familiar with the molecular biology of archaea.

We moved the description of archaeal transcription at the beginning of the introduction implementing this part with further information to improve comprehension for readers not familiar with transcription and archaea.

While this review focuses on RIP and TFS4, it would be extremely interesting to know whether RNAP inhibitors/negative regulators have been found in other archaeal viruses/archaea.

At lines 157-165, we added information about the current knowledge of RNAP/transcription repressors in archaea.

Figures: Please use “simple colors” rather than “sea green” and explain what is on the figs ( i.e., rudder) and the color scheme for each;  in Fig.2 B, eTFIIB looks yellow to me;  

Reviewer #2 seems to agree about the usage of sea green which is commonly used elsewhere and is not simple green, while we have changed khaki in yellow in Fig.2 B. We described the figure content in the captions (i.e. rudder) of all figures. 

in Figs 2 and 3, colored ions are not visible for the most part;

Ions that are not clearly visible in the figure have been removed from the figure legend.

in Fig. 4, it is difficult to see the different colors for Myxopyronin and Fidaxomicin.

We have changed Fig. 4 to avoid confusion about Myxopyronin and Fidaxomicin.

Specific comments:

L 50: remove one “reached”

Removed. Now line 117.

L54-56: awkward sentence; please rewrite

Rewrote to make it more comprehensible. Now lines 122-124.

L79: give a bit more info about MAF1; this comes out of the blue after the sentence on HIV.

Additional information added. Now lines 146-149.

L83-86: moves “,” so it is easier to understand, i.e., “The archaeal transcription cycle is closely related to RNAPII that, in addition to the RNAPII subunits includes bona fide homologues of general transcription factors TBP, TFIIB, TFIIE, TFIIS, Spt4/5, Elf1, and CPSF73…

Increasing the description of the archaeal transcription cycle, this part has been completely rewritten and moved at the beginning of the introduction.

Reviewer 2 Report

The authors described the global transcription regulation through multisubunit RNA polymerase (RNAP) inhibition occurring by either its own transcription factors or virus infection in Archaea. These repression mechanisms can be used for the development of new antibiotics.

Specifically, this review focused on the structural mechanisms of conserved characteristics of RNAP inhibition by viral RIP and cellular regulator TFS4. The manuscript is well written.

Minor comments:

-Figure 1, What are the TSS and TTS? Transcription start and termination sites? It can be spelled out in the figure legend.

-In figure 2, the TFIIB is shown in yellow, not in khaki. RIP (in sea green) needs to be annotated in b as well and where is the clamp in the B?

-Reference 49: no page number/doi information is shown for this reference.

-Line 219, Ref 58 can be replaced with references that directly describe the p53 or myc with RNAPII.

-Line 218: “cis-platin” to cis-platinum or cisplatin.

Author Response

Minor comments:

-Figure 1, What are the TSS and TTS? Transcription start and termination sites? It can be spelled out in the figure legend.

We specified TSS and TTS abbreviation meaning in the caption of the figure.

-In figure 2, the TFIIB is shown in yellow, not in khaki. RIP (in sea green) needs to be annotated in b as well and where is the clamp in the B?

Color changed accordingly and RIP annotated correctly. Figure 2 has been changed to annotate the position of the clamp in panel B as requested.

-Reference 49: no page number/doi information is shown for this reference.

Corrected. Now Ref50.

-Line 219, Ref 58 can be replaced with references that directly describe the p53 or myc with RNAPII.

Ref 59 replaced with other two references specific about p53 (Ref 69) and myc (Ref 70).

-Line 218: “cis-platin” to cis-platinum or cisplatin.

Corrected with cis-platinum. Now line 322.

Reviewer 3 Report

Although the topic is not in my area of expertise, I found this to be a very well-written, engaging and interesting review of the field of archael RNAP inhibition. The figures are very clear and the authors make well-justified analogies. They have also been comprehensive with the referencing, although I'm not sure I would know if there are any missing references. In conclusion, this is a thoughtful review that will be very useful to the field and will be well cited. 

Author Response

Although the topic is not in my area of expertise, I found this to be a very well-written, engaging and interesting review of the field of archaeal RNAP inhibition. The figures are very clear and the authors make well-justified analogies. They have also been comprehensive with the referencing, although I'm not sure I would know if there are any missing references. In conclusion, this is a thoughtful review that will be very useful to the field and will be well cited.

We thank the reviewer for the positive comments.